# The Impact of Cesarean and Vaginal Delivery on Results of Psychological Cognitive Test in 5 Year Old Children

**DOI:** 10.3390/medicina56100554

**Published:** 2020-10-21

**Authors:** Barbora Blazkova, Anna Pastorkova, Ivo Solansky, Milos Veleminsky, Milos Veleminsky, Andrea Rossnerova, Katerina Honkova, Pavel Rossner, Radim J. Sram

**Affiliations:** 1Faculty of Health and Social Sciences, University of South Bohemia, 370 05 Ceske Budejovice, Czech Republic; barbora.blazkova01@gmail.com (B.B.); PastorkovaA@seznam.cz (A.P.); solansky.ivo@seznam.cz (I.S.); mveleminsky@tbn.cz (M.V.); 2Institute of Experimental Medicine CAS, Videnska 1083, 142 20 Prague, Czech Republic; andrea.rossnerova@iem.cas.cz (A.R.); katerina.honkova@iem.cas.cz (K.H.); pavel.rossner@iem.cas.cz (P.R.J.); 3Hospital Ceske Budejovice, a.s., 370 01 Ceske Budejovice, Czech Republic

**Keywords:** vaginal delivery, cesarean section, psychological development, cognitive development, Bender Visual Motor Gestalt Test, Raven Colored Progressive Matrices

## Abstract

*Background and objectives:* The impact of cesarean and vaginal delivery on cognitive development was analyzed in 5 year old children. *Materials and Methods*: Two cohorts of 5 year old children born in the years 2013 and 2014 in Karvina (Northern Moravia) and Ceske Budejovice (Southern Bohemia) were studied for their cognitive development related to vaginal (*n* = 117) and cesarean types of delivery (*n* = 51). The Bender Visual Motor Gestalt Test (BG test) and the Raven Colored Progressive Matrices (RCPM test) were used as psychological tests. *Results*: In the comparison of vaginal delivery vs. cesarean section, the children delivered by cesarean section scored lower and, therefore, achieved poorer performance in cognitive tests compared to those born by vaginal delivery, as shown in the RCPM (*p* < 0.001) and in the BG test (*p* < 0.001). When mothers’ education level was considered, the children whose mothers achieved a university degree scored higher in both the RCPM test (*p* < 0.001) and the BG test (*p* < 0.01) compared to the children of mothers with lower secondary education. When comparing mothers with a university degree to those with higher secondary education, there was a significant correlation between level of education and score achieved in the RCPM test (*p* < 0.001), but not in the BG test. *Conclusions*: According to our findings, the mode of delivery seems to have a significant influence on performance in psychological cognitive tests in 5 year old children in favor of those who were born by vaginal delivery. Since cesarean-born children scored notably below vaginally born children, it appears possible that cesarean delivery may have a convincingly adverse effect on children’s further cognitive development.

## 1. Introduction

In recent years, a possible impact of the type of delivery on the neurobehavioral functions of children has been discussed. Despite the recommendation of the World Health Organization suggesting that the average percentage of cesarean section deliveries should not be higher than 15% of all deliveries, the rate in developed countries has now reached now 25% [1]. According to a multi-country cross-sectional study [2], cesarean section represents one of the most frequently performed surgical procedures in the world.

Cesarean delivery requested by mothers without any maternal or fetal indication seems to be rising [3]. As a surgical procedure, the cesarean section may be related to some child health adversities such as allergies, asthma, diabetes mellitus, and celiac disease, as already found [4].

So far, there is little knowledge regarding the known effects of the mode of delivery on neurodevelopmental trajectories in children. Measured intelligence is perceived as a strong predictor of further personal development in life such as physical and mental health and mortality [5].

Khadem and Khadivzadeh [6] in Iran studied the changes in intelligence quotient (IQ) in two matched groups of 189 children aged 6–7 years delivered either by cesarean section (CS) or by spontaneous vaginal delivery (VD). They did not observe any association between the type of delivery and cognitive development of those children.

In the Longitudinal Study of Australian Children, Polidano et al. [7] measured differences in child cognitive performance at 4–9 years of age between cesarean-born and vaginally born children (*n* = 3666). The authors found that, across several measures, only cesarean-born children at the age of 8–9 performed significantly below vaginally born children, by up to one-tenth of a standard deviation in national numeracy test scores.

Between December 2007 and March 2008, Khalaf et al. [8] investigated in Ireland the effect of spontaneous VD (*n* = 43,927) vs. elective CS (*n* = 9460). Parents evaluated their children’s behavioral, cognitive, and motor developmental outcomes at the age of 9 months using the Ages and Stages Questionnaire (ASQ), and at the age of 3 years using the Strengths and Difficulties Questionnaire (SDQ). At the age of 9 months, elective CS affected scores for personal social skills (odds ratio (OR) = 1.41, 95% confidence interval (CI) 1.19–1.66, *p* < 0.05), problem solving (OR = 1.27, 95% CI 1.06–1.53, *p* < 0.05), and gross motor function (OR = 1.71, 95% CI 1.44–2.02, *p* < 0.05). At the age of 3 years, no increased risk in SDQ scores was observed. According to the conclusions from this study, children who were born by elective cesarean section might face a higher risk of neurodevelopmental delay.

In Australia, Smithers et al. [9] analyzed the school assessment of children at the age of 8 years by comparing cesarean section (*n* = 650) vs. vaginal birth (*n* = 2959) in women with a previous cesarean delivery. The assessment included reading, writing, spelling, grammar, and numeracy. Analyses suggest that previous cesarean section did not increase the risk of poor school achievement of children at the age of 8 years.

Curran et al. [10] assessed school performance in Swedish adolescents of ages 14–21, using the data from the birth register from 1982–1995 (*n* = 1,489,925), with 1,036,424 children born by unassisted vaginal delivery (VD) and 42,107 by elective cesarean section (CS). Poor school performance was observed in 138,202 unassisted VD vs. 7064 elective CS, OR = 1.06 (95% CI 1.03–1.09). Therefore, this study did find an association between delivery by CS and school performance.

The impact of the two different modes of delivery on neuropsychological development was also investigated by Gonzales-Valenzuela et al. [11] who compared vaginal (*n* = 84) and cesarean delivery (*n* = 40) in 6 year old children born in twin births. Children were evaluated in their first year of primary school, using the Child Neuropsychological Maturity Questionnaire and Kaufman’s Intelligence Test. Significant differences were found for verbal development, nonverbal development, global development, and general intelligence. Cesarean delivery was found as a possible risk factor for neuropsychological development and intelligence in twin births.

Zhou et al. [12] measured the development of cesarean (*n* = 401) and vaginally delivered (*n* = 1354) children at the age of 1–59 months using the ASQ. They did not observe any risk in the ASQ domains of communication, fine motor, gross motor, problem solving, personal, and social. Their findings suggest that cesarean delivery does not increase or decrease the risk of developmental delay in children when compared to vaginal delivery.

While some of these studies indicated an impact of cesarean delivery on cognitive development in children, other studies did not observe any difference between vaginal and cesarean deliveries. Nevertheless, according to a systematic review and meta-analysis performed by Zhang et al. [13], birth by cesarean delivery might be associated with certain neurodevelopmental and psychiatric disorders. Therefore, we tested the question whether there was a significant correlation between the mode of delivery and performance in cognitive tests in 5 year old children in our cohorts.

The children were selected from our cohorts in two districts which differ by the level of air pollution: the higher polluted district of Karvina (Northern Moravia) and the control district of Ceske Budejovice (Southern Bohemia) [14].

## 2. Materials and Methods

### 2.1. Subjects

The cohorts were created in the summer of 2013 and winter of 2014 from newborns born at the Ceske Budejovice Hospital, Department of Obstetrics and Gynecology and Department of Neonatology, and at the Karvina Hospital, Department of Obstetrics and Gynecology and Department of Neonatology. Newborns were selected from full-term pregnancies with physiologic course (38th–41st week) of nonsmoking mothers who signed a written consent form. Cohorts included 99 newborns (summer) and 100 newborns (winter) in Ceske Budejovice and 71 newborns (summer) and 74 newborns (winter) in Karvina. Vaginal deliveries did not include any instrumental deliveries (the frequency of vacuum extraction (VEX) was around 2%, while that of forceps was around 0.9%; therefore, they were not observed due to the size of our cohorts). Cesarean sections were mostly planned. Indications for Cesarian sections were as follows: breech presentation of the fetus when mother did not accept vaginal delivery (approximately 40%), elective repeat C-section (ERCS) in mothers with previous C-section in cases when mother did not accept VBAC (vaginal birth after cesarean section) or there was another medical indication for the C-section (rigidity of birth canal, large baby) (approximately 40%), cephalopelvic disproportion (CPD) (approximately 10%), and other reasons such as primary neurologic indication, primary ophthalmologic indication, or primary orthopedic indication (approximately 10%).

The study was approved by the Ethics Committee of both hospitals and the Institute of Experimental Medicine CAS in Prague.

Between November 2018 and November 2019, 199 mothers from the Ceske Budejovice district and 143 from the Karvina district who provided blood and urine samples from their newborns in 2013 and 2014 were approached to take part in psychological testing. Out of the total number of 342 potential subjects, 5 years after joining the study, 140 refused to take part in the psychological examination and 31 were impossible to contact. In the present study, the data were collected from 99 children from Ceske Budejovice and 70 children from Karvina. The presented study, therefore, includes 169 children. The data collected included a questionnaire, nonverbal intelligence test, and visual motor functioning test. The relationship between the type of delivery and psychological tests was analyzed in these children.

This study was approved by the Ethical Committee of the Faculty of Health and Social Sciences, University of South Bohemia, on 30 June 2017.

### 2.2. Questionnaire for Mothers

We asked mothers who were engaged in our study to fill out a questionnaire that consisted of 11 categories: personal data, social environment, mother’s level of education (lower secondary, higher secondary, and university as terms used in the Czech Republic), cigarette smoking, breastfeeding, the child’s kindergarten attendance, allergic diseases of the tested child, symptoms of a disease of tested child within the last year, eating habits, presence of domestic animals, and household conditions. When analyzing the results, we examined information gathered in relation to the psychological test results.

Breastfeeding was adopted as a potential confounder from the pediatric questionnaire in form of the full breastfeeding in months, and the continuous parameter was transformed into a binary scale using a cutoff point of six months.

Father and sibling interactions were not evaluated.

### 2.3. Measures of Child Visual Motor Functioning and Intellect

The Bender Visual Motor Gestalt Test (BG test) [15] is a psychological tool assessing visual motor functioning for screening potential developmental disorders, delays, or impairments. This test is usually presented to test subjects of 3 years of age and older (including adults). The BG test consists of nine cards depicting various geometric shapes. This type of drawing test was chosen for its easy administration and the fact that it focuses on cognitive abilities such as visual acuity, visual perception, motor functioning, and perceptual abilities. Since the test requires drawing and does not take a long time to complete, it also provided a convenient way of engaging 5 year old children in the test situation and helped them to adapt and stay motivated to perform during the test situation.

In the present study, a psychologist presented each figure in sequence to the test subject and asked the subject to copy it onto a sheet of paper, while asking the child to make the best copy possible. This test has no time limit and its administration usually took around 10 min. After the test completion, the results were scored on the basis of accuracy and organization. In total, 168 children at the age of 5 years completed the test.

After completing the Bender Visual Motor Gestalt Test (BG test), the children were assessed by the culture-free nonverbal Raven Colored Progressive Matrices (RCPM test) [16], which is widely used for children between the ages of 5 and 11 years. It is a simple, fast, and easy-to-administer test that can be used even for persons with physical or mental impairment. The RCPM test is aimed at measuring so-called fluid intelligence, reasoning, and problem-solving ability, which are believed to be the core components of general intelligence. This multiple-choice test consists of 35 items arranged in increasing levels of difficulty. The test is not timed. The children were presented with a series of patterns with parts missing and asked to decide which piece was missing from the presented options. A total of 167 children at the age of 5 years completed the RCPM test.

According to our experience, 5 year old children are not always ready to complete complex IQ tests upon their first visit. Therefore, more sessions are often required so that the child adapts to the test situation and successfully completes the test. The given tests (the RCPM test and the BG test) were chosen bearing in mind the fact that they are usually well accepted by 5 year old children which would facilitate adaptation to the test situation. Both the psychometric qualities and the time effectiveness of the chosen tests were considered as significant advantages, and mothers were found to be more willing to take part in the study. Out of all tested children, only one was not able to undergo the assessment, and one child was only able to complete the BG test. Both tests were administered by a trained psychologist in a quiet place provided by the hospitals in Ceske Budejovice and in Karvina. The psychologist was blinded to the type of delivery (as well as all other information gathered by the questionnaire). The exact age at the day of measurement was calculated by from the difference between the date of birth and the date of psychological testing. When collecting the data, the education level of mothers was also taken into consideration.

### 2.4. Statistical Analysis

We used two statistical methods for the evaluation of differences in the cohorts. The Mann–Whitney U-test (Wilcoxon rank-sum test) was used for direct comparison of the RCPM test and BG test results between cohorts.

Logistic regression was used for the purpose of estimating the impact of the type of delivery on the scores of the RCPM test and BG test as dependent values. A necessary conversion of rough scores of the test values into binary scale was done by dividing medians of the appropriate group distribution. Logistic regression was used for the quantification of impact intensity to calculate the odds ratio (OR), thus estimating the strength of the association between vaginal delivery and cesarean delivery when achieving a testing score above the median of the group distribution.

The odds ratio is expressed as the ratio of the odds of event A in the presence of event B and the odds of event A in the absence of event B. An OR equal to 1 suggests no association, an OR above 1 denotes a positive association, and an OR below 1 denotes a negative association of events. The OR value also quantifies how often an event is associated [17]. Calculated ORs in this analysis described the probability of children achieving scores above the median in their cohorts in association with the type of delivery and mothers’ education in both the RCPM test and the BG test.

To be able to exclude all other possible confounders of estimated impacts, multiple other parameters were tested with respect to health and social status of mothers, mostly related to the maternal questionnaire, such as maternal age, maternal ETS (environmental tobacco smoke), various maternal health status parameters, children birth parameters and birth procedures, and quantified child illness by categories in the period from birth to 2 years of age. No other statistically significant impact was found this way.

## 3. Results

The results of psychological tests related to the type of delivery are listed in Table 1. Lower results and, therefore, poorer performance in cognitive tests used in the comparison of vaginal delivery vs. cesarean section could be seen in cesarean-born children, in both the RCPM test (*p* < 0.001) and the BG test (*p* < 0.001). When we compared both types of delivery in the less polluted Ceske Budejovice, vaginally born children achieved better results in both tests (*p* < 0.05), while, in the more polluted Karvina, a higher level of differences was observed in the RCPM test (*p* < 0.01) compared to the BG test (*p* < 0.05). When we analyzed this effect according to gender, statistically significant differences between vaginal type and cesarean section were seen in boys in both the RCPM test (*p* < 0.001) and the BG test (*p* < 0.001). Those differences were not statistically significant in girls.

In addition, we also considered mothers’ education level. In the comparison of mothers at university level to those at the lower secondary level, the RCPM test detected statistically significantly better results in all children (*p* < 0.001) born to mothers with higher education. Similar results were observed for the subsets of Ceske Budejovice (*p* < 0.01), Karvina (*p* < 0.001), boys (*p* < 0.001), and girls (*p* < 0.001). In a similar comparison using the BG test, better results were detected in all children (*p* < 0.01), as well as in the subsets of Karvina (*p* < 0.01) and boys (*p* < 0.05), but not in the Ceske Budejovice group and in girls. Comparing university level vs. higher secondary education, a higher score in test performance was only achieved in the RCPM test in all children (*p* < 0.001), as well as in the subsets of Ceske Budejovice (*p* < 0.05), Karvina (*p* < 0.01), boys (*p* < 0.05), and girls (*p* < 0.05) (Table 2).

The impact of breastfeeding was also analyzed. Initially, only the simple “breastfeeding—yes/no” parameter was assessed. Since the number of non-breastfeeding mothers was too small, no significant impact on the psychological test results was observed. Since the parameter mentioned above was found to be insufficient for a more detailed analysis, we adopted another two parameters from the questionnaires: full breastfeeding and partial breastfeeding lengths (both in months). We split the breastfeeding length values into a binary scale of less or more than 6 months. Non-breastfeeding mothers were assigned to the first group (less than 6 months).

Cross-checking tests showed that there were no statistically significant associations between the type of delivery and breastfeeding; however, a strong positive association between breastfeeding and maternal university degree was found.

The results for the vaginal type of delivery vs. cesarean section, mothers’ university education vs. other education levels, and breastfeeding above 6 months vs. below were slightly different using the RCPM and the BG tests (Table 3). The logistic regression showed a significant impact of vaginal type of delivery and mothers’ university education in a model used for estimating the above-median RCPM test results for both cohorts (OR 2.32, *p* < 0.01) and a highly significant impact of mothers’ university education in the Karvina and boys groups (OR 6.60 with *p* < 0.001 and OR 4.55 with *p* < 0.01, respectively). On the contrary, multivariate analysis showed that the above-median BG test results were significantly more dependent on the delivery type in both cohorts and for boys (OR 1.99 and 3.00 with *p* < 0.05, respectively).

The analysis of the simultaneous impact on psychological tests using the multivariate model of the same factors did not show any substantial change in the results related to the type of delivery and breastfeeding length (Table 4); thus, these parameters could be considered as independent of each other.

Results obtained from the maternal questionnaires are presented in Table 5. Child ETS exposure during the first and second years of age was higher in Karvina vs. Ceske Budejovice (0.14 ± 0.35 vs. 0.03 ± 0.17 cigarettes (cig)/day, *p* < 0.01 and 0.19 ± 0.39 vs. 0.06 ± 0.24, *p* < 0.01, respectively). Several confounders significantly differed in Karvina vs. Ceske Budejovice; the gestation age was longer (40.1 ± 2.00 vs. 39.5 ± 39.5 ± 1.5 weeks, *p* < 0.001), the birth length was shorter (49.2 ± 2.2 vs. 50.0 ± 1.9 cm, *p* < 0.01), the Apgar 5′ score was higher (10.0 ± 0.1 vs. 9.8 ± 0.6, *p* < 0.001), and the TBC primovaccination was higher (12.7 ± 33.5 vs. 3.9 ± 19.4%, *p* < 0.05). The incidence of some children diseases was also higher in Karvina, such as GIS (gastrointestinal system) (0.44 ± 0.75 vs. 0.21 ± 0.55, *p* < 0.010) and viral diseases (0.73 ± 1.06 vs. 0.05 ± 0.26, *p* < 0.001). However, those tested confounders did not affect the results of both psychological tests.

## 4. Discussion

We tested the hypothesis of whether there was a significant correlation between the mode of delivery and the performance in cognitive tests in 5 year old children. In our study, we observed a very significant positive impact of the vaginal type of delivery on the psychological cognitive test results in 5 year old children, which was more pronounced in boys. The observed results were not affected by any of the tested confounders. These outcomes correspond to those from studies by Polidano et al. [7], Curran et al. [10], and Gonzales-Valenzuela et al. [11]. Potential effects of the type of delivery on cognitive development in children should also be followed in our cohorts in the future to see if the observed changes are long-lasting or if they disappear at a later age.

The IQ score of children, assessed on the basis of the performance in the intelligence test, correlated positively with the level of maternal education. These findings are in accordance with the common paradigm regarding the heritability of general cognitive abilities [18], which seems to increase from a low value in early childhood of about 30% to well over 50% at adulthood and continuing into old age [19]. Recent genome-wide association studies have successfully identified inherited genome sequence differences that account for 20% of the 50% heritability of total intelligence [20].

The mechanisms of the negative impact of cesarean section on children’s neurobehavioral functions may be explained by several factors. When a child is born by cesarean section, the child is not exposed to the mother’s vaginal microbiota at birth [21]. Changes in the infant microbiome contribute to changes in children’s metabolic pathways [22]. These differences in microbiota result in a different postnatal development of the immune system [23]. Experimental studies described the changes in dopamine concentrations and changes in dopamine-mediated behaviors [24], as well as noradrenaline concentrations [25]. It is postulated that this difference in microbiota may affect the development of brain and the later increase in the risk of diseases such as schizophrenia [26]. The differences in the type of delivery and children’s cognitive development related to the effect of the gut microbiota may be explained by chemical signaling from the gut microbiota to the central nervous system, affecting memory, motivation, mood, and reactivity to stress [27].

Another possible explanation for the differences in test performance could have been the result of the influence of the environment. Even though we included many parameters such as sex, socioeconomic parameters, morbidity, air pollution, and others, there remain other variables such as father and sibling interactions, childhood diet, way of upbringing, and social enrichment [18]. Other variables, difficult to detect and quantify, may yet have a great impact on neurodevelopment with long-term effects stronger than the mode of birth. The possible impact of air pollution on psychological development in our cohorts will be further analyzed.

A limitation of our research was the total number of participants. By presenting this evidence, we would like to motivate more research within this topic. A larger number of participants, as well as methods and diagnostic tests, may help us to understand these problems in further detail.

According to our findings, birth by cesarean section may present a higher risk related to psychological development of children compared to vaginal birth, especially in boys. The cesarean section is one of the most frequently performed surgical procedures in the world, and the rates of the cesarean delivery seem to be rising. We believe there is a need for a better understanding of all the potential effects and impacts of this method of delivery to gain more detailed insight into an informed decision-making process in order to be able to consider all the benefits and risks of each type of delivery.

## 5. Conclusions

We analyzed the impact of cesarean delivery and vaginal delivery on the cognitive development of 5 year old children, using the Bender Visual Motor Gestalt Test (BG test) and the Raven Colored Progressive Matrices (RCPM test). Children born by cesarean section achieved poorer performance in both tests compared to children born by vaginal delivery. This effect was more pronounced in boys. The results of cognitive tests were also influenced by mothers’ education level. Our findings support the hypothesis that cesarean section may have an adverse effect on children’s further cognitive development.

## Figures and Tables

**Table 1 medicina-56-00554-t001:** Results of psychological tests and the type of delivery. CB, Ceske Budejovice; RCPM, Raven Colored Progressive Matrices; BG, Bender Visual Motor Gestalt.

Type of Delivery	All		CB		Karvina		Boys		Girls	
Vaginal	*N*	Mean ± SD	*N*	Mean ± SD	*N*	Mean ± SD	*N*	Mean ± SD	*N*	Mean ± SD
RCPM test	116	19.4 ± 4.4 ***	69	19.5 ± 4.2 *	47	19.3 ± 4.9 **	54	19.8 ± 4.3 ***	62	19.2 ± 4.6
BG test	117	34.7 ± 14.4 ***	69	34.8 ± 14.5 *	48	34.7 ± 14.4 *	55	35.3 ± 15.2 ***	62	34.3 ± 13.8
Cesarean section	*N*	Mean ± SD	*N*	Mean ± SD	*N*	Mean ± SD	*N*	Mean ± SD	*N*	Mean ± SD
RCPM test	51	16.9 ± 4.5	30	17.6 ± 4.2	21	15.9 ± 4.8	24	16.3 ± 4.6	27	17.4 ± 4.4
BG test	51	27.3 ± 15.7	30	28.2 ± 13.6	21	26.0 ± 18.5	24	23.9 ± 16.5	27	30.3 ± 14.5

Results of Mann–Whitney U-test comparing type of delivery: * *p* < 0.05; ** *p* < 0.01; *** *p* < 0.001.

**Table 2 medicina-56-00554-t002:** Results of psychological tests and the mothers’ education level.

Mothers’ Education Level										
	All		CB		Karvina		Boys		Girls	
Lower secondary	*N*	Mean ± SD	*N*	Mean ± SD	*N*	Mean ± SD	*N*	Mean ± SD	*N*	Mean ± SD
RCPM test	16	15.0 ± 2.7 ***	5	15.6 ± 2.9 **	11	14.7 ± 2.8 ***	9	15.0 ± 3.4 ***	7	15.0 ± 1.8 ***
BG test	17	24.1 ± 12.9 **	5	25.0 ± 15.7	12	23.8 ± 12.2 **	10	23.7 ± 15.0 *	7	24.7 ± 10.2
Higher secondary	*N*	Mean ± SD	*N*	Mean ± SD	*N*	Mean ± SD	*N*	Mean ± SD	*N*	Mean ± SD
RCPM test	51	18.1 ± 4.4 ***	30	18.4 ± 4.3 *	21	17.4 ± 4.7 **	24	18.0 ± 5.0 *	27	18.1 ± 4.1 *
BG test	51	31.8 ± 14.3	30	32.4 ± 14.3	21	30.5 ± 14.6	24	29.9 ± 17.4	27	32.9 ± 12.1
University	*N*	Mean ± SD	*N*	Mean ± SD	*N*	Mean ± SD	*N*	Mean ± SD	*N*	Mean ± SD
RCPM test	51	20.6 ± 4.5	30	20.4 ± 3.9	21	20.7 ± 5.2	24	20.5 ± 4.0	27	20.6 ± 5.1
BG test	51	35.2 ± 16.0	30	34.5 ± 14.7	21	36.1 ± 17.6	24	35.1 ± 14.5	27	35.3 ± 17.8

Results of Mann–Whitney U-test comparing type of mothers’ education: * *p* < 0.05; ** *p* < 0.01; *** *p* < 0.001.

**Table 3 medicina-56-00554-t003:** Bivariate impact of the delivery type, mothers’ education level, and breastfeeding length on psychological testing values. OR, odds ratio; CI, confidence intervals; CS, cesarean section.

		All	CB	Karvina	Boys	Girls
		OR (CI)	OR (CI)	OR (CI)	OR (CI)	OR (CI)
RCPM test	Vaginal delivery vs. CS	2.32 ** (1.21–4.47)	1.98 (0.84–4.67)	2.86 (1.02–8.05)	2.88 * (1.10–7.53)	1.91 (0.77–4.71)
	University vs. other education	2.82 *** (1.45–5.51)	1.76 (0.78–3.98)	6.60 *** (2.19–19.85)	4.55 ** (1.71–12.10)	1.87 (0.74–4.73)
	Full breastfeeding ≥6 months	2.58 *** (1.39–4.81)	2.71 ** (1.22–6.05)	2.74 * (1.04–7.23)	5.67 *** (2.18–14.73)	1.40 (0.60–3.26)
BG test	Vaginal delivery vs. CS	1.99 * (1.03–3.82)	2.05 (0.87–4.83)	2.00 (0.73–5.51)	3.00 * (1.12–8.05)	1.39 (0.57–3.39)
	University vs. other education	1.14 (0.62–2.11)	0.83 (0.37–1.85)	2.65 (0.99–7.11)	1.63 (0.67–3.95)	0.82 (0.34–1.95)
	Full breastfeeding length ≥6 months	3.10 *** (1.67–5.76)	4.41 *** (1.92–10.12)	3.05 ** (1.15–8.09)	4.88 *** (1.89–12.59)	2.25 (0.97–5.20)

Bivariate logistic regression results: * *p* < 0.05; ** *p* < 0.01; *** *p* < 0.001.

**Table 4 medicina-56-00554-t004:** Multivariate impact of the delivery type, mothers’ education level, and breastfeeding length on psychological testing values.

		All	CB	Karvina	Boys	Girls
		OR (CI)	OR (CI)	OR (CI)	OR (CI)	OR (CI)
RCPM test	Vaginal delivery vs. CS	2.16 ** (1.09–4.28)	1.76 (0.73–4.28)	2.67 (0.85–8.43)	2.90 (0.95–8.81)	1.92 (0.77–4.81)
	University vs. other education	2.11 * (1.03–4.31)	1.30 (0.55–3.12)	4.88 ** (1.51–15.82)	2.60 (0.89–7.56)	1.76 (0.66–4.67)
	Full breastfeeding ≥6 months	2.01 * (1.03–3.91)	2.39 * (1.03–5.55)	1.81 (0.58–5.63)	4.84 *** (1.69–13.85)	1.08 (0.44–2.67)
BG test	Vaginal delivery vs. CS	1.99 * (1.00–3.96)	1.90 (0.76–4.76)	1.92 (0.64–5.75)	3.57 ** (1.20–10.62)	1.28 (0.51–3.22)
	University vs. other education	0.67 (0.34–1.36)	0.41 (0.15–1.08)	1.67 (0.56–4.99)	0.75 (0.26–2.15)	0.57 (0.22–1.50)
	Full breastfeeding length ≥6 months	3.59 *** (1.83–7.06)	5.78 *** (2.24–14.90)	2.92 * (1.00–8.50)	5.82 *** (2.01–16.83)	2.74 * (1.10–6.81)

Multivariate logistic regression results: * *p* < 0.05; ** *p* < 0.01; *** *p* < 0.001.

**Table 5 medicina-56-00554-t005:** Overview of tested confounders. ETS, environmental tobacco smoke; GIS, gastrointestinal system.

		All		CB		Karvina		Boys		Girls	
		*N*	Mean ± SD	*N*	Mean ± SD	*N*	Mean ± SD	*N*	Mean ± SD	*N*	Mean ± SD
**Maternal Characteristics**											
Maternal age	years	174	31.9 ± 4.5	103	32.6 ± 4.4	71	30.8 ± 4.4	82	31.5 ± 4.0	92	32.3 ± 4.9
ETS—pregnancy	%	173	5.8 ± 23.4	103	3.9 ± 19.4	70	8.6 ± 28.2	82	6.1 ± 24.1	91	5.5 ± 22.9
ETS—1st child year	%	173	7.5 ± 26.4	103	2.9 ± 16.9	70	14.3 ± 35.2 **	82	8.5 ± 28.1	91	6.6 ± 25.0
ETS—2nd child year	%	173	11.0 ± 31.4	103	5.8 ± 23.5	70	18.6 ± 39.2 **	82	11.0 ± 31.5	91	11.0 ± 31.4
**Birth Characteristics**											
Gestation age	weeks	174	39.8 ± 1.7	103	39.5 ± 1.5	71	40.1 ± 2.0 ***	82	39.6 ± 1.3	92	39.9 ± 2.0
Birth weight	g	168	3436 ± 442	101	3471 ± 460	67	3385 ± 412	80	3507 ± 437	88	3372 ± 440 ^++^
Birth length	cm	165	49.7 ± 2.1	98	50.0 ± 1.9	67	49.2 ± 2.2 **	80	50.1 ± 2.1	85	49.3 ± 2.0 ^++^
Birth head perimeter	cm	164	34.4 ± 1.4	101	34.5 ± 1.5	63	34.4 ± 1.3	78	34.7 ± 1.4	86	34.2 ± 1.4 ^++^
Apgar 5′		151	9.9 ± 0.5	92	9.8 ± 0.6	59	10.0 ± 0.1 ***	69	9.8 ± 0.6	82	9.9 ± 0.3
Other delivery complication	%	174	5.7 ± 23.3	103	5.8 ± 23.5	71	5.6 ± 23.2	82	8.5 ± 28.1	92	3.3 ± 17.9
Hyperbilirubinemia	%	174	9.2 ± 29.0	103	5.8 ± 23.5	71	14.1 ± 35.0	82	8.5 ± 28.1	92	9.8 ± 29.9
TBC primovaccination	%	174	7.5 ± 26.4	103	3.9 ± 19.4	71	12.7 ± 33.5 *	82	4.9 ± 21.7	92	9.8 ± 29.9
Breastfeeding	%	174	96.0 ± 19.7	103	95.1 ± 21.5	71	97.2 ± 16.7	82	95.1 ± 21.7	92	96.7 ± 17.8
Full breastfeeding	months	170	4.6 ± 2.5	102	4.7 ± 2.5	68	4.51 ± 2.6	80	4.7 ± 2.6	90	4.6 ± 2.5
Partial breastfeeding	months	173	11.2 ± 7.7	102	11.3 ± 7.3	71	11.1 ± 8.2	82	10.7 ± 7.7	91	11.7 ± 7.6
**Children’s Diseases**											
GIS	count	174	0.30 ± 0.65	103	0.21 ± 0.55	71	0.44 ± 0.75 **	82	0.33 ± 0.74	92	0.28 ± 0.56
Viral diseases	count	174	0.33 ± 0.78	103	0.05 ± 0.26	71	0.73 ± 1.06 ***	82	0.27 ± 0.69	92	0.38 ± 0.85
Otitis	count	174	0.28 ± 0.56	103	0.22 ± 0.52	71	0.37 ± 0.62	82	0.28 ± 0.53	92	0.28 ± 0.60
HCD	count	174	0.13 ± 0.42	103	0.17 ± 0.49	71	0.07 ± 0.31	82	0.13 ± 0.38	92	0.12 ± 0.47
Bronchitis	count	174	0.01 ± 0.08	103	0.00 ± 0.00	71	0.01 ± 0.12	82	0.00 ± 0.00	92	0.01 ± 0.10

Results of Mann–Whitney U-test comparing region: * *p* < 0.05; ** *p* < 0.01; *** *p* < 0.001; results of Mann–Whitney U-test comparing gender: ^++^
*p* < 0.01.

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
