# Peer review of "The Impact of Cesarean and Vaginal Delivery on Results of Psychological Cognitive Test in 5 Year Old Children"

_medicina, 2020, doi:10.3390/medicina56100554_

Round 1

Reviewer 1 Report

Some important concerns regarding the methods still remain unsolved.

According to author´s statements, participants were selected from normal deliveries (38th-41st week) of 103 nonsmoking mothers who signed a written consent, but the authors do not contribute a definition for normal deliveries, and some cesarean deliveries have been also included. This is a major concern. Authors do not provide data about the indication for the cesarean deliveries (fetal distress, maternal conditions, malpresentations, dystocia...) and it is crucial to support their conclusions, because some of these pre-existing variables could have affected children development. 

Author Response

Reply to Reviewer 1

Thank you for your comments.

  • The text was corrected:

Newborns were selected from full term pregnacies with physiologic course (38th-41st week) of nonsmoking mothers who signed a written consent.

  • Cesarean sections were mostly planned. To the text was added:

Indications for Cesarian sections were: Breech presentation of the fetus when mother did not accept vaginal delivery  (approx.. 40 %),  Elective repeat C-section (ERCS) in mothers with previous C-section in cases, when mother didn't accept VBAC (vaginal birth after cesarean section) or there was another medical indication for the C-section (rigidity of birth canal, large baby) (approx.. 40 %), Cephalopelvic disproportion (CPD) (approx.. 10 %), other reasons as primary neurologic indication, primary ophthalmologic indication, primary orthopedic indication (approx.. 10 %).  

  • To the text was added:

Vaginal deliveries did not include any instrumental deliveries (the frequency of VEX is around 2 %, forceps around 0.9 % - therefore they were not observed due to the size of our cohorts). Cesarean sections were mostly planned. 

Reviewer 2 Report

I still have issues with the paper 

This is a secondary analysis, of a study on pollution, so not powered, nor designed. Consequently, this should be read as an opportunistic additional analysis

Secondly, there is a major ‘fall out’ in the retention, as only 196 cases were analysed. Were there differences between in included and excluded cases. We need this analysis. 

Finally, caeasarean is analysed, but were these preplanned, or urgent caesareans, and why have instrumental vaginal deliveries been excluded ? In the references provided and discussed in the introduction, it seems that elective caesareans were compared to vaginal delivery. In another attempt to rephrase this, were data analysed by intention to treat (intended vaginal) or as treated. In the second setting, this is another bias, even more as instrumental deliveries were not included.

Although my knowledge on the Czech language is limited, I understood that the last question was not understood by the authors (how big is the claimed effect); This can be extracted by tables 1 and 2

Author Response

Reply to Reviewer 2

Thank you for your comments.

  • It is true that study was originally planned to analyze the impact of PAHs pollution on cognitive functions. The impact of the type of delivery was analyzed after results observed by Polidano et al. [7], Khalaf et al. [8], Curran et al. [10], Zhou et al. [12], Zhang et al. [13] to check the idea about the impact of the type of delivery on cognitive functions in children.
  • We could analyze only 169 cases. It was very difficult to contact mothers five years after delivery, 140 refused to take part in the psychological examination and 31 were impossible to contact.

In attached table is displayed characteristic‘s comparision of cases included with psychological testing and the others without unachievable psychological test values

  • To the text was added:

Cesarean sections were mostly planned. Indications for Cesarian sections were: Breech presentation of the fetus when mother did not accept vaginal delivery  (approx.. 40 %),  Elective repeat C-section (ERCS) in mothers with previous C-section in cases, when mother didn't accept VBAC (vaginal birth after cesarean section) or there was another medical indication for the C-section (rigidity of birth canal, large baby) (approx.. 40 %), Cephalopelvic disproportion (CPD) (approx.. 10 %), other reasons as primary neurologic indication, primary ophthalmologic indication, primary orthopedic indication (approx.. 10 %).  

  • To the text was added:

Vaginal deliveries did not include any instrumental deliveries (the frequency of VEX is around 2 %, forceps around 0.9 % - therefore they were not observed due to the size of our cohorts).

Table: Overview of tested confounders

All

CB

Karvina

Boys

Girls

N

Mean±SD

N

Mean±SD

N

Mean±SD

N

Mean±SD

N

Mean±SD

Maternal Characteristics

Maternal Age

years

with tests

174

  31.9±4.5 *)

103

32.6±4.4

71

  30.8±4.4 *)

82

31.5±4.0

92

32.3±4.9

cohort remainder

182

30.7±4.6

106

31.9±4.1

74

29.1±4.9

86

30.9±4.4

91

30.5±4.8

ETS - pregnancy

%

with tests

173

5.8±23.4

103

3.9±19.4

70

8.6±28.2

82

6.1±24.1

91

5.5±22.9

cohort remainder

126

11.9±32.5

76

6.6±25.0

48

20.8±41.0

62

11.3±31.9

62

12.9±33.8

ETS - 1st child year

%

with tests

173

7.5±26.4

103

2.9±16.9

70

14.3±35.2

82

8.5±28.1

91

6.6±25.0

cohort remainder

127

13.4±34.2

76

6.6±25.0

49

24.5±43.5

62

11.3±31.9

63

15.9±36.8

ETS - 2nd child year

%

with tests

173

11.0±31.4

103

5.8±23.5

70

18.6±39.2

82

11.0±31.5

91

11.0±31.4

cohort remainder

128

14.1±34.9

77

9.1±28.9

49

22.4±42.2

63

12.7±33.6

63

15.9±36.8

Birth Characteristics

Gestation Age

weeks

with tests

174

39.8±1.7

103

39.5±1.5

71

40.1±2.0

82

39.6±1.3

92

39.9±2.0

cohort remainder

129

39.67±1.29

76

39.49±1.43

51

40.02±0.95

64

39.69±1.50

63

39.67±1.06

Birth Weight

g

with tests

168

3436±442

101

3471±460

67

3385±412

80

3507±437

88

3372±440

cohort remainder

181

3439±442

105

3474±427

74

3416±439

86

3501±428

92

3378±452

Birth Length

cm

with tests

165

49.7±2.1

98

50.0±1.9

67

49.2±2.2

80

50.1±2.1

85

49.3±2.0

cohort remainder

173

49.7±1.8

97

50.0±1.5

74

49.4±2.0

83

50.0±1.9

87

49.3±1.6

Birth Head Perimeter

cm

with tests

164

34.4±1.4

101

34.5±1.5

63

34.4±1.3

78

34.7±1.4

86

34.2±1.4

cohort remainder

167

34.7±1.7

99

34.7±1.3

66

34.8±2.2

81

34.9±1.3

83

34.6±2.0

Apgar 5'

with tests

151

9.9±0.5

92

9.8±0.6

59

10.0±0.1

69

9.8±0.6

82

9.9±0.3

cohort remainder

162

9.8±0.5

93

9.7±0.6

67

10.0±0.1

80

9.9±0.4

80

9.8±0.5

Other Delivery Complication

%

with tests

174

  5.7±23.3 *)

103

5.8±23.5

71

5.6±23.2

82

   8.5±28.1 *)

92

3.3±17.9

cohort remainder

128

12.5±33.2

75

10.7±31.1

51

15.7±36.7

62

9.7±29.8

64

15.6±36.6

Hyperbilirubinia

%

with tests

174

9.2±29.0

103

5.8±23.5

71

  14.1±35.0 *)

82

8.5±28.1

92

9.8±29.9

cohort remainder

130

6.2±24.1

77

7.8±27.0

51

3.9±19.6

64

7.8±27.1

64

4.7±21.3

TBC Primovaccination

%

with tests

174

7.5±26.4

103

3.9±19.4

71

12.7±33.5

82

4.9±21.7

92

9.8±29.9

cohort remainder

126

3.2±17.6

74

1.4±11.6

50

6.0±24.0

64

1.6±12.5

60

5.0±22.0

Breast Feeding

%

with tests

174

96.0±19.7

103

95.1±21.5

71

97.2±16.7

82

95.1±21.7

92

96.7±17.8

cohort remainder

131

96.2±19.2

78

96.2±19.4

51

96.1±19.6

65

93.8±24.2

64

98.4±12.5

Full Breast Feeding

months

with tests

170

4.6±2.5

102

4.7±2.5

68

4.51±2.6

80

4.7±2.6

90

4.6±2.5

cohort remainder

131

4.6±2.3

78

4.8±2.1

51

4.0±2.4

65

4.4±2.5

64

4.7±2.1

Partial Breast Feeding

months

with tests

173

11.2±7.7

102

11.3±7.3

71

11.1±8.2

82

10.7±7.7

91

11.7±7.6

cohort remainder

131

10.4±7.4

78

11.0±7.1

51

8.9±7.3

65

9.7±7.5

64

10.9±7.2

Children's Diseases

GIS

count

with tests

174

0.30±0.65

103

0.21±0.55

71

0.44±0.75

82

0.33±0.74

92

0.28±0.56

cohort remainder

130

0.29±0.45

79

0.23±0.42

51

0.37±0.49

66

0.29±0.46

62

0.29±0.46

Viral Diseases

count

with tests

174

0.33±0.78

103

0.05±0.26

71

0.73±1.06

82

0.27±0.69

92

0.38±0.85

cohort remainder

130

0.19±0.39

79

0.01±0.11

51

0.45±0.50

66

0.14±0.35

62

0.24±0.43

Otitis

count

with tests

174

0.28±0.56

103

0.22±0.52

71

0.37±0.62

82

0.28±0.53

92

0.28±0.60

cohort remainder

130

0.13±0.34

79

0.14±0.35

51

0.12±0.33

66

0.17±0.38

62

0.08±0.27

HCD

count

with tests

174

0.13±0.42

103

0.17±0.49

71

0.07±0.31

82

0.13±0.38

92

  0.12±0.47 *)

cohort remainder

130

0.82±0.39

79

0.80±0.40

51

0.84±0.37

66

0.89±0.31

62

0.74±0.44

Bronchitis

count

with tests

174

0.01±0.08

103

0.00±0.00

71

0.01±0.12

82

0.00±0.00

92

0.01±0.10

cohort remainder

130

0.30±0.46

79

0.32±0.47

51

0.28±0.45

66

0.36±0.48

62

0.24±0.43

Results of Mann Whitney U-test compare between included and excluded cases *) p ~ 0.05,   **) p ~ 0.01,  ***) p ~ 0.001

Round 2

Reviewer 1 Report

The paper has been improved.

The authors have included the indications for the c-sections.

I have some concerns regarding the management of the covariables and confounding factors, and perhaps it should be included as a limitation of the study

Reviewer 2 Report

none